# Subnational tailoring of malaria interventions for strategic planning and prioritization: Experience and perspectives of five malaria programs

Letitia Onyango[1], Ghislaine Ouédraogo-Ametchie[1], Ifeoma Ozodiegwu[1,2¤], Beatriz Galatas[3], Jaline Gerardin[1*]

1 Institute for Global Health and Department of Preventive Medicine, Northwestern University Feinberg School of Medicine, Chicago, Illinois, United States of America, 2 Department of Health Informatics and Data Science, Loyola University Parkinson School of Health Sciences and Public Health, Chicago, Illinois, United States of America, 3 Global Malaria Programme, World Health Organization, Geneva, Switzerland

¤ Current Address: Department of Health Informatics and Data Science, Loyola University Parkinson School of Health Sciences and Public Health, Chicago, Illinois, United States of America
* jgerardin@northwestern.edu

**Data availability statement:** Given the small sample, and identifiability of participants, the full transcripts cannot be provided. Codebooks and aggregated, anonymized analyses will

## Abstract

In the context of high malaria burden and insufficient resources, several national malaria programs (NMPs) used subnational tailoring (SNT) as a tool for evidence-informed decision-making on their national malaria strategic plans and funding requests. SNT is a data-informed method of developing intervention plans that maximize health impact under limited resources across heterogenous areas and populations. The SNT process included the formation of an SNT team, determination of criteria for targeting interventions, data assembly and review, stratification, application of targeting criteria to determine preliminary plans, mathematical modeling, finalization of intervention plans, and monitoring and evaluation of the eventual implemented plan, all under the leadership of the NMP. Analysis steps of SNT were supported by the World Health Organization (WHO) and other partners. As SNT was a new approach, this study used semi-structured interviews to understand the perspectives and experiences of personnel from five NMPs (Burkina Faso, Ghana, Guinea, Nigeria, and Togo) that undertook SNT between 2019 and 2023. Participants reported that SNT outputs were used to inform national strategic plans and prioritized plans, that the process incentivized improvements in data collection and data quality, and that NMPs were strongly motivated to grow their capacity to conduct more steps of the SNT analysis process internally. Major challenges included the lack of resources available to implement the full strategic plans as well as challenges with data quality and alignment of stakeholders. Participants reported a moderate to strong sense of ownership over the process and were eager to extend, adapt, and reuse the SNT process in the future. Among countries supported by WHO, SNT was well-accepted and allowed NMPs to successfully use evidence to inform their decision-making, advocate for themselves, and mobilize resources.

be available upon request. Requests for data should be made to the Northwestern University Institutional Review Board at irb@northwestern.edu.

**Funding:** LO, GAO, and JG were funded by grants from the Bill & Melinda Gates Foundation to JG: INV-002092 and INV-048410. The funder had no role in study design, data collection and analysis, decision to publish, or preparation of the manuscript.

**Competing interests:** The authors have declared that no competing interests exist.

## Introduction

After concerted efforts to increase access to vector control [1], chemoprevention [2], and prompt diagnosis and treatment [1], malaria incidence worldwide declined by 27% between 2000 and 2015 [3]. However, progress slowed after 2015. The World Malaria Reports of 2017 and 2018 showed that global progress towards the 2020 targets set out in the World Health Organization's (WHO) Global Technical Strategy for Malaria was unlikely to be achieved [4,5].

Stalled progress was driven by trends in countries with high malaria burden, including the 11 countries (Burkina Faso, Cameroon, Democratic Republic of the Congo, Ghana, Mali, Mozambique, Niger, Nigeria, Uganda, United Republic of Tanzania, and India) that accounted for 70% of the global estimated case burden and 71% of the estimated deaths. In the WHO African Region, malaria case incidence and malaria mortality have remained flat since 2018 [6]. Many factors contribute to rising or stable malaria incidence in high-burden countries, including the underlying intensity of malaria transmission, socio-demographic and epidemiologic risk factors, poor access to care, sub-optimal malaria intervention coverage including in hard-to-reach areas, poor implementation quality, rising resistance to insecticides and antimalarials, and stagnation in global funding for malaria [5].

In November 2018, WHO and the RBM Partnership to End Malaria launched the High Burden to High Impact (HBHI) response to get the highest burden countries back on track to achieve the 2025 milestones set out in WHO's Global Technical Strategy for malaria [7]. The HBHI approach had four key response elements: political will to reduce malaria deaths; strategic information to drive impact; better guidance, policies, and strategies; and a coordinated national malaria response. Between 2018 and 2020, the HBHI approach was implemented in the 10 highest-burden African countries [8,9].

The purpose of HBHI response element 2 – use of strategic information to drive impact – was to address the need by national malaria programs (NMPs) and technical partners for context-specific analysis [7]. Under HBHI response element 2, local data and quantitative methods are used to identify challenges for progress against malaria and develop appropriate responses to achieve impact.

Since 2019, many countries have used elements of HBHI response element 2 to review their national malaria strategic plans (NMSPs) and submit funding requests to the Global Fund to Fight Aids, Tuberculosis and Malaria. Under the lead of NMPs, and in partnership with WHO and analysis partners, the following areas of work were implemented to improve the use of strategic information and achieve maximum impact in their NMSPs and funding requests:

- **Establishment of national malaria data repositories**: NMPs worked with their country's health management and information system to establish functioning national malaria data repositories with program tracking dashboards.

- **Implementation of progress reviews** consisting of country-level malaria situation analysis and review of malaria programs to understand progress and bottlenecks.

- **Subnational tailoring of interventions (SNT)** to develop final strategic and prioritized plans with more efficient targeting of resources.

In SNT, intervention strategies are tailored to individual subnational units, accounting for subnational heterogeneity in malaria risk and other local variation in determinants of malaria transmission such as seasonality, vector behavior, and access to health services [6,10]. SNT moves away from a one-size-fits-all approach to intervention targeting [11] and instead provides a process by which effectiveness and cost-effectiveness of intervention strategies can be optimized under a limited resource envelope.

The SNT process recommended by WHO includes these steps:

1) **Assembly of an SNT team** led by the NMP, consisting of local, regional, and/or global partners in addition to the NMP itself.

2) **Determination of criteria** to be used to target interventions under consideration in the national strategic plan.

3) **Collection of relevant data** and **stratification of indicators** required for decision-making, including epidemiological stratification and stratification of determinants of malaria transmission.

4) **Geographical targeting of interventions** based on the defined criteria, stratification maps, and any relevant operational constraints, to prepare targeted intervention mixes for NMSPs and prioritized plans for funding requests.

5) **Mathematical modeling** to evaluate the potential impact of different intervention mix scenarios posed by the NMP in the previous step.

6) **Consensus** reached on the final strategic intervention mix per subnational area and **costing** of the national strategic plan.

7) **Prioritization of interventions** to maximize impact if resources are insufficient to fully cover the NMSP. Steps 2–6 are repeated using the costed strategic plan as the basis and considering aspects such as operational feasibility, equity, acceptability, and/or cost-effectiveness.

8) **Monitoring and evaluation** of the impact of the operationally costed plan to optimize its effectiveness and ensure maximum impact will be reached.

In 2019–2021 (Phase 1), all 10 original African highest-burden countries and several additional ones, including Guinea, conducted SNT as part of their HBHI response element 2 activities to finalize their NMSPs and applications to the Global Fund. Technical portions of the SNT analysis were led by WHO in nearly all countries and supported by various external analytical groups, as NMPs in Phase 1 were less familiar with stratification and the use of mathematical modeling to inform decision-making.

In 2022–2023 (Phase 2), at least 28 countries used some or all elements of SNT analysis to inform their strategic planning and/or applications to the Global Fund or to Gavi, the Vaccine Alliance [10]. During Phase 2, most but not all countries coordinated their SNT activities with support from WHO. Technical portions of SNT were executed directly by the country, WHO, or other external analysis partners, depending on the country. In countries where WHO was not engaged in Phase 2, the steps of the SNT process sometimes deviated from the abovementioned steps.

This study uses the term "analysis partner" to refer to an external group that executes analytical steps of the SNT process. In some instances, particularly during Phase 1, WHO participated as an analysis partner, but this was separate to its role as a coordinator of the SNT process. In-person meetings were held with all Phase 1 countries in 2019 to launch and implement the SNT steps. However, due to the COVID-19 pandemic, most engagements between NMPs and partners became fully remote at the end of Phase 1, and engagements remained nearly always remote during Phase 2.

The implementation of SNT brought with it substantial innovation and change to the previous processes used by most countries for decision-making. While the technical outputs of applications of SNT or SNT-like analyses have already been documented [9,12–19], understanding the NMP experience with the SNT process is necessary to identify successful elements, remaining gaps, and approaches that need adjustment, such that countries become self-reliant in the use of data to inform priority-setting.

This study evaluates the perceptions of NMPs toward the SNT analysis activities that they undertook in Phase 1 and/or Phase 2, focusing on NMPs' motivations for initiating SNT; how SNT outputs were used; challenges with the process and use of outputs, including challenges with partners; sense of ownership of the process; and future plans for SNT.

## Methods

### Study design

This study used a qualitative approach based on semi-structured interviews with a purposive sample of participants. Given the nuances of the themes of the study, the multi-phased aspects SNT implementation, and the impact of country dynamics of SNT initiation and progression, an interview approach offered participants the opportunity to speak in-depth about their experiences and paint a robust picture of SNT implementation in their countries. Semi-structured interviews provided the flexibility for participants to discuss their experiences with SNT in terms of their country context and roles, while focusing on aspects of SNT implementation closest to their experience. This enabled an inductive approach to analysis, and a deeper exploration of how NMP expectations and relationships with analysis partners affected observed results.

### Study population and participant recruitment

Study participants were recruited between 1 September 2022 and 31 July 2023 from NMPs in countries that had undergone an SNT exercise in Phase 1 and/or 2 with technical support from WHO and had implemented at least steps 1–6 of the SNT process described above. NMPs included were Burkina Faso, Ghana, Guinea, and Nigeria, that used SNT in 2019–2021 (Phase 1) and in 2022–2023 (Phase 2); and Togo, whose first SNT experience was in Phase 2.

NMP managers were contacted by email from WHO inviting participation in the study. NMP managers who expressed interest were subsequently contacted by the Northwestern University research team (LO or GAO), and NMP managers nominated potential NMP representatives for interviews. Interviewers (LO and GAO) did not have prior relationships with participants. The study team drafted letters outlining the parameters of the study and a link to provide online consent via Qualtrics. Study participants were informed that interviewers were neutral and separate from the WHO SNT team or analysis partner teams. After participants completed their online consent, interviewers then led to recruitment follow-up, which included emails and text messages to schedule the interview. The consent form is provided in S1 File.

All NMP representatives nominated by their NMP manager were contacted by the study team, and all responsive representatives were interviewed. Two or three representatives were interviewed from each NMP, resulting in a total of 12 participants. NMP representatives included the NMP manager, the SNT focal person within the NMCP, and/or data manager, depending on the country. All study participants had direct involvement in the SNT analysis process and attended meetings with WHO and analysis partners during SNT. Participants led the coordination of SNT activities including strategic plan development, data management, and communication with analysis partners. Given the purposive nature of the sample and limited number of individuals within each NMP working on SNT, two to three participants at each site were sufficient for achieving saturation.

### Data collection

Interviewers developed the initial interview guide in consultation with an evaluation team member from WHO with experience supporting SNT. The interview guide was piloted with volunteers who were experienced with SNT implementation in

the study countries but outside the evaluation team, including another WHO member and two previous NMP managers. Pilot interviews were recorded, transcribed, and closely reviewed to identify redundancies and improve question clarity. The final interview guide is provided in S2 File.

At the beginning of each interview, interviewers reviewed the consent language contained in the written consent document. Participants were informed that their responses were confidential, including from WHO members of the study team; that narrative results would be reported at the country level; and that any quotes would be completely anonymized. Interviews were conducted in person (5) or virtually (7), in English or French, by LO or GAO. Interviews took place between June 2023 and October 2023 and lasted between 45 and 60 minutes. For in-person interviews, interviewers traveled to conferences and workshops where participants indicated they would be present. Virtual interviews were conducted over Zoom. All interviews were audio recorded and supplemented with written notes. Interviewers sought verbal consent from participants to record the interviews.

Interview recordings were transcribed by the transcription company Multilingual Connections. Transcriptions were subsequently quality-assessed against the original audio by the research team (LO and JG). Transcriptions of French interviews were translated into English using the software DeepL. English translations were reviewed next to the original French by bilingual members of the research team (LO, GAO, and JG) to ensure accurate translation transcript content, key acronyms, and malaria terminology.

### Data analysis

The study team developed an initial coding architecture based on research questions. The study team then chose three transcripts to review, using an interpretive phenomenological analysis approach [20], to identify additional codes and ensure that participant experiences were centered throughout. This approach enabled the research team to assess how participants defined their experiences with SNT and their relationships with analysis partners.

LO and JG analyzed the remaining transcripts and adapted the codebook until all phenomena were sufficiently represented. The codebook was finalized after two transcripts from each country were complete, resulting in a final codebook with both deductive and inductive codes. Given the small sample size, all coding and thematic analysis was completed in Excel, resulting in two workbooks: one focused on country-level dynamics and another outlining cross-country themes in alignment with research questions.

### Ethical review

This study was reviewed and approved by Northwestern University's Institutional Review Board (project STU00215997). The World Health Organization Research Ethics Review Committee granted this study an ERC exemption (project HBHI-SOC). Participants provided written and oral informed consent to participate.

## Results

### Initiating the SNT analysis process

Participants cited multiple motivations for their NMP's initiation of and participation in the SNT process. NMPs that engaged in multiple rounds of SNT reported that during Phase 1, they were primarily driven by a push from funders and WHO. Participants in two countries said that the push from the Global Fund to refine their strategic plans with stratification was an important factor in initiating the SNT process. For other countries, HBHI designation in Phase 1 drove the political will to explore alternative approaches, including SNT, to malaria control and elimination.

Armed with knowledge and experience from Phase 1, NMPs reported more self-driven motivations in Phase 2. NMPs were more focused on how the SNT analysis process could enable them to develop better strategic plans, identify priority

areas for specific interventions, and stretch operational budgets. Participants cited a growth in confidence and successful partnerships with analysis partners in Phase 1 as additional motivators during Phase 2:

*We had a first phase where we were recommended by our Global Fund partner to stratify our activities and identify the most relevant ones. [...] That's when we really got in touch with modeling. And then there's a second phase which was just recently when we wrote our strategic plan and applied for funding from the Global Fund. Since we had completed the first phase, which enabled us to see the usefulness of modeling, we thought it would be appropriate to use stratification. [...] I think the time we were most confident when we knew what we were doing, was the second time. Because we had some idea about modeling. We were leading the process this time. And we were more informed.* (Participant 7)

During both phases, internal discussions, strategic plan performance reviews, and the 2018 World Malaria Report catalyzed NMPs to reconsider their previous approaches and to integrate stratification into national strategic plans and funding applications. In the face of dwindling resources for malaria interventions, participants recognized the limitations of previous "one size fits all" approaches and the need to strategize resource allocation to improve intervention effectiveness and have a bigger impact on malaria burden or to progress toward elimination:

*We said to ourselves: we're tired of doing the same things everywhere at the same time, because everything we do is the same in every district. And we really said to ourselves, we need to look and see if there are districts that are in pre-elimination or moderate transmission, for example, that we can push towards pre-elimination or elimination. [...] We need to direct resources to where they're needed. That way, we can really boost our objectives. Because today, resources are becoming increasingly scarce.* (Participant 11)

When asked about their expectations of the SNT process, participants expressed that they expected to be able to understand their malaria burden at the district level and make intervention plans appropriate to each district, thereby increasing the overall impact of the country's malaria strategy:

*Our expectation was to be able to show the situation by district. This will enable us to see what action we can take in this district and what efforts we can make in other districts. So, just to be able to really target interventions to increase their impact.* (Participant 12)

*My expectations were, foremost, to be able to make choices based on the different forecasts considering the interventions that we were going to put in place.* (Participant 1)

Some participants were concerned with whether analysis outputs would generate feasible recommendations or meet NMP needs, but were open to trying the process:

*Our expectation was that, okay, let's see what they want to do because it was a new concept to everyone. And we felt with the level of control the country had with bed nets, let's see how, even with the bed nets, we would be able to stratify what kind of bed nets go everywhere.* (Participant 8)

## Use of SNT outputs

Participants in all countries provided examples of how SNT outputs supported internal efforts to revise strategic plans, prioritize interventions, and implement more thorough malaria program reviews. Participants also discussed how SNT outputs enabled them to identify limitations in their data and advocate for NMP activities and priorities more effectively with stakeholders.

**Motivating improvement of data availability and quality**

The SNT experience motivated participants from all NMPs to improve data quality and availability. All participants recognized the importance of high-quality data for decision-making:

> *It's important that before you embark [and] pull data for decision making, analysis and modeling and subnational tailoring, you must have the data. So, it's important to improve the quality of data that's available. And so, there's effort being put into that to ensure that we have data to make decisions [...] because you cannot work with nothing.* (Participant 10)

Participants reported that the data mobilization phase was initially challenging, especially when NMPs relied on other Ministry of Health units for surveillance and health facility-level data. Several participants described their NMP's use of the SNT process to identify limitations and inefficiencies in their data collection:

> *For the SNT process, the backbone has always been garbage in, garbage out. [...] The main challenge has always been getting the good data, or the complete data, because our surveillance has less sensitivity in terms of getting the right data from the places where the services have been delivered. If we get the right data into the whole SNT process, improve data quality, improve coverage of the data we get and get it at the right time, then it will now give us a better output of what we do. We could do better in getting the right data into the pool.* (Participant 9)

Challenges with data quality and availability helped NMPs identify interdependencies with other ministries and subnational health offices, from whom it was occasionally challenging to acquire timely data. For example, NMPs incorporating insecticide resistance data needed additional input from the Ministry of Agriculture, while NMPs incorporating health facility visit data relied on health facilities for up-to-date data. These challenges resulted in delayed timelines for SNT implementation:

> *There were some data that the team required but was not readily available. Particularly data on vector species distribution. And largely the entomology surveillance data. We didn't have all of that at [the program]. We had written to the Center for Medical Research where they were the ones implementing that on behalf of [the program], and we didn't get any response. [...] Definitely that also impacted the timeline.* (Participant 8)

Participants reported that data quality reviews were a valuable learning experience in understanding how to leverage the full capabilities of SNT and mathematical modeling. Limited data quality and granularity affected NMPs' ability to get the most out of the SNT process at every stage, from stratification to mathematical modeling, and occasionally required certain compromises:

> *There was some data that we couldn't put into the model, even if it was important because the quality wasn't reliable. I'll just take the example of the data on estimated [incidence]. We had to add the population's ability to use health facilities in the event of fever because to be tested, you must go to a health facility. This indicator was so bad that every time we included it, it biased all the results. So we didn't include it in the [incidence] estimate, even though it was important.* (Participant 7)

Participants reported that NMPs responded to data availability and quality challenges encountered during SNT by defining NMSP priorities for data quality improvement, developing national malaria data repositories, and implementing new data documentation policies, among other activities.

## Strategic planning

All participants reported that SNT outputs were used in strategic planning, prioritization, or both.

*Everything we do now is based on stratification. For each intervention in different regions, results were based on stratification. Therefore, the stratification results have been inserted in the current revised strategic plan. For example, we know that [in each region] this is what is done, based on the stratification results.* (Participant 2)

*Every time we initiated some activity, we tried to justify it by referring to the recommendations in this [SNT] report as we implemented it.* (Participant 7)

Several participants reported that SNT enabled the identification of areas eligible for receiving seasonal malaria chemoprevention (SMC) that were not previously receiving SMC. SNT also identified SMC-eligible areas that should be receiving additional cycles of SMC based on the length of their transmission season and allowed NMPs to explore the possible impact of expanding SMC to children above the age of 5. When asked about the impact of SNT on funding applications, one NMP reported that their use of SNT outputs enabled them to make bolder, data-informed funding requests for expanding SMC:

*It [SNT] helped us because we were already discussing the number of [SMC] cycles with the Global Fund. We went with four. They accepted that. So we thought - are they going to accept that we go to five cycles? Because according to the rainfall data review, rainfall in the southern SMC zone exceeds four months. In some places, it's as much as six months. But if it's four cycles [of SMC], that means there are two months that aren't covered. With the stratification, there was evidence [that] showed that this is it. So nobody can say no, this isn't it. That made it easier for us.* (Participant 12)

All participants reported using SNT to identify priority areas for distribution of second-generation long-lasting insecticide-treated nets (LLINs) through review of local data from coverage surveys, entomological surveillance, and insecticide resistance surveillance. For one NMP, the SNT process revealed that pyrethroid-only LLINs were likely still highly effective in some districts. For other NMPs, SNT analysis provided evidence in support of switching to second-generation LLINs:

*We started using second-generation LLINs for certain regions, certain districts, and ordinary LLINs for certain regions because of the vectors' resistance to insecticides. The stratification process helped us to clarify all these factors.* (Participant 2)

## Intervention prioritization and preparation of funding requests

Participants stated that SNT helped them better manage their resources and improve the effectiveness of their malaria strategies by targeting resources appropriately to priority areas, such as the deployment of a limited-availability malaria vaccine:

*We were able to determine the priorities, which districts have the highest priority based on the burden. They are the districts we chose to vaccinate.* (Participant 1)

*Before, we simply divided the country into three parts according to the seasonal landscape. […] As a result [of SNT], our priorities were geared more towards where the stratification had enabled us to put our finger on the difficulties in these regions.* (Participant 3)

NMPs reported using the results of SNT when preparing their funding applications to the Global Fund to justify the epidemiological impact of, and the districts targeted with, their proposed interventions. Since SNT until this point had focused on effectiveness rather than cost-effectiveness, in some cases, further prioritization was necessary to account for costs:

*Some activities, even if they were relevant according to [SNT], but considering the cost were put aside or were adjusted. [...] The latest generation of LLINs, we were only able to get them for one locality [in Phase 1] because it was impossible to include all the districts identified by stratification. [...] When the second stratification was done, which showed practically the same thing, the need for [second-generation LLINs], in the funding request we justified that we didn't want to have the previous mosquito net type, we wanted to have [second-generation LLINs] available at least in such-and-such districts in significant quantities.* (Participant 7)

*We also had the opportunity to apply for funding from the Global Fund. [SNT] helped us a lot because when we got there, we were able to think about it. We went for targeted interventions. Before, we didn't do that. [...] When we wanted to present the results of the funding request to the Global Fund, we put the modeling results. [...] We've shown that if we really have the money to implement all that, these are the lives we can save. [...] And thus when we presented the system, there was no question of [Global Fund] sincerely validating what we wanted to do.* (Participant 11)

## Engagement with funders and partners

Some participants expressed that before undertaking SNT, their NMP was initially concerned about the acceptability of SNT results by their communities or partners:

*We were also a bit skeptical that when these analytical results eventually come back to the country, what will people do with those results? You know different funders have different [areas of] focus. It's possible that the result comes back, and people say, I'm not interested in the result. It is based on what the American people have signed with us that we do.* (Participant 8)

Despite pre-SNT concerns, participants in all countries described how SNT outputs were critical in streamlining funding applications, garnering buy-in from funders and communicating with country partners. In some countries, participants felt that the inclusion of SNT outputs in their funding application gave their NMP greater legitimacy:

*[SNT] gets stakeholders to buy into the seriousness of the program. I think one of the key things is how it encourages funders to understand that this program is really serious, and they really want to be efficient with the use of the resources.* (Participant 4)

Participants felt that this added layer of legitimacy helped them use SNT as an advocacy tool when communicating with funders, implementing partners, and other stakeholders. Participants reported having to often provide justification for their activities to NMP partners whose buy-in was dependent on well-articulated plans and justifications. Participants expressed that SNT outputs like stratification maps and intervention impact predictions allowed them to develop language for communicating their goals with external stakeholders, including prospective donors and implementing partners, and more effectively garner buy-in:

*The most important lesson for me is that stratification has enabled me [...] to make a case to an authority. Even when you say that my incidence is twelve per 1000 [they will say] "I don't see that." But when you present the map and say - "Well, here we are, where we are today, we were like that, we were in the red. Now we're in yellow, we want to go*

*green" for example…So it's easy to use that for them, to make a case, to explain. Do you see, then, that when you have figures like that, that you throw at people, it doesn't say anything? That's true. So, it's really important. It's a tool, it's a tool for analysis. It's also an advocacy tool.* (Participant 11)

*It became very easy to communicate to partners about where and why I am deploying this intervention here. So that process was really paramount. We couldn't have just said we have an elimination strategy without showing that we have gone through this process.* (Participant 5)

*When we talked to the Global Fund about what we had planned, the fact that we had used this stratification and the mathematical projection meant that we could easily refer to this stratification to make ourselves better understood by the Global Fund in terms of the efficiency of the interventions we were proposing.* (Participant 3)

**Early perceived impact of SNT-informed decision-making on malaria burden**

Participants reported early results of how intervention plans designed with SNT reduced malaria burden and intervention implementation. Participants from one NMP reported that SNT outputs helped identify higher than expected burden in a specific region. The NMP chose to distribute second-generation LLINs in that area, resulting in significant reduction in prevalence:

*When we distributed new generational nets in that year, the recent demographic and health survey showed a significant reduction in prevalence. […] The only difference we had done or intervention we had put in there is a new generation of nets, that was based on the stratification and knowing that this place is high burden. So stratification and subnational tailoring is so important.* (Participant 5)

Participants from another NMP reported that because of revised intervention targeting efforts after SNT, some districts moved from high transmission rates to moderate or low transmission rates. Participants from a third NMP described how SNT outputs helped to mobilize funding for increased SMC, resulting in a significant reduction in malaria deaths over the last few years.

**Challenges with the SNT process and use of outputs**

Participants identified limitations in data quality (discussed above) and financial resources as the largest challenges they faced when undergoing the SNT process and using its outputs, respectively. Participants cited additional challenges related to coordinating SNT-related decisions with external stakeholders and working with analysis partners.

**Challenges with using SNT outputs**

Participants were not able to implement the entirety of their evidence-informed strategy due to resource constraints:

*It's funding that poses a problem every time [...] For example, we have indoor residual spraying which will not be implemented, we have prevention, SMC, which we would like to extend to nine-year-olds which will not be implemented because of lack of resources.* (Participant 1)

All NMPs reported that they were actively working to identify solutions to funding gaps that would allow them to expand interventions to all eligible populations and areas defined during the SNT process. For a few NMPs, implementation of indoor residual spraying (IRS) was especially challenged by the absence of resources and/or differing funder priorities:

*We can use IRS to reduce the number of cases, but we don't have the resources. […] [The funder] said, well, they won't do the IRS, because in the first year they came to finance our gaps. They can't do the IRS when we have gaps for mosquito nets, we have gaps for medicines and so on. They prefer to finance our gaps and carry out feasibility studies for the IRS. […] We've agreed that in 2024, we'll do the research.* (Participant 11)

Some NMPs explored the implications of implementing recommended interventions at reduced capacity. In these cases, SNT helped NMPs understand to what extent implementing an intervention at reduced coverage would help them move towards the goals articulated in their strategic plan.

Some recommendations from SNT were challenging for NMPs to implement on their own. Participants from one NMP reported that recommendations to increase antenatal consultations and increase intermittent preventive treatment for malaria in pregnancy were only possible with funding to overhaul the health system, which did not feel feasible at the time.

### Challenges with in-country partner management

NMPs were responsible for determining which internal and external stakeholders to involve in the SNT process. Many NMPs chose to establish technical working groups or steering committees for implementation, forming SNT teams which included international non-governmental organizations (NGOs), local NGOs, and in one case, a local university. NMPs were also reliant on national statistics agencies, the Ministry of Health, and the Ministry of Agriculture to support data mobilization for SNT.

Two NMPs reported challenges with partner coordination in relation to competing partner priorities. These participants reported tension between implementing partners, analysis partners, and the NMP with regard to SNT. For one NMP, implementers did not agree with the NMP's choice to change coverage in specific areas after review during SNT. For another NMP, disagreements about prevalence estimation led to tense discussions between the NMP, analysis partners, and implementing partners. In the face of these challenges, however, participants reported that the NMP maintained their autonomy through continued engagement and compromise and identified solutions that aligned with their priorities even when key stakeholders were not supportive of SNT:

*Some of these stakeholders did not understand why we needed to do stratification. [...] There was a lot of pushback with some of the implementers saying, look, the reality on ground is this is not feasible at this time, it's still too premature to go this route. So there's still a lot of engagement and compromising here and there, things that need to see how everyone can be carried along and try to understand how we can do it better. A lot of back and forth. We did make them understand that that's the direction the program wants to go because we have funding constraints. [...] Some say we don't have sufficient data for now to begin to do things like that. But the message was passed.* (Participant 10)

In addition to resistance from implementing partners, participants across all but one NMP reported challenges with the acceptability of outputs from interministerial and subnational health office partners. In these instances, subnational offices were initially resistant to recommendations emerging from SNT outputs. Participants attributed this resistance to poor communication about SNT from the national level to the subnational level, and a failure to involve subnational offices from the very beginning. In the absence of a collective understanding of why decisions were being made, subnational offices could feel alienated or unsupported:

*Of the 21 [districts], only 16 are eligible for SMC. A normal human would be thinking, why are you excluding the other five? And the question is germane because when you begin to move towards the subnational level, things become more political than scientific or professional. They feel [that] because they didn't support the government, is that why you excluded them? Why don't you just allow the whole place to be eligible for something?* (Participant 8)

As NMPs began to improve their communication about SNT data inputs and the role of SNT in improving operational efficiency, they experienced greater cooperation and acceptability of results. However, NMPs recognized that there is still a long way to go with improving interministerial and subnational coordination:

*It's important to empower the field staff. By focusing on that, it's already helping them to better understand the whole process that leads to the output of these data. If we don't, we can't have a similar understanding of the power and precision [of SNT]. (Participant 3)*

*In the beginning, […] when we wanted to go to these targeted interventions, people didn't understand right away where we wanted to go. But when we really explained and showed the point of doing this, people were interested, and then we all got excited about achieving our goal. And that's why, when we had finished the results, we organized a national workshop where we invited all the regional directors, all the players in the country, where we presented all the stages of what we had done, and people were interested. (Participant 12)*

**Challenges with analysis partners**

In all countries, WHO provided overall guidance for stratification and data centralization. In two NMPs, the stratification and intervention targeting portions of the SNT analysis were implemented by an internal staff member, whereas in the other three NMPs, remote external analysis partners executed these tasks. All NMPs relied on external partners for mathematical modeling.

Despite positive relationships with analysis partners, participants described several operational challenges with the logistics of partner management. Most participants felt that the time they initially allocated for SNT analysis was not enough, given partner commitments and concurrent activities within the NMP:

*We were under pressure in terms of time, and that's when we were a bit worried. But in the end, they [analysis partners] were quite reactive and that enabled us to catch up. In fact, I think it's just a matter of communicating, and I know they didn't just have our job to do either, so it wasn't always easy for them to be available all the time. (Participant 3)*

For most NMPs, communication with external partners was completely remote and compromised by poor connections and short meetings. Even though NMPs held internal discussions about SNT outputs after each analysis presentation, they felt it would have been more beneficial to have analysis partners participating in those conversations. Some participants felt that in-person collaboration or visits would have streamlined discussions, created more clarity about the analysis process, and enabled partners to learn more about the inner workings of the NMP that may affect SNT analysis:

*The other thing that bothered us was that we couldn't work face-to-face. Everything was done virtually. Whereas if we'd been face-to-face, we might have benefited from other support. If the team had come to [country], they could have seen how our data collection system works. We could also learn a few lessons over there, to keep improving quality. We've worked hard on this whole process, we've worked remotely. [...] If we want to improve, at least working face-to-face, it will allow the country to acquire other important elements as well. (Participant 12)*

The two NMPs whose stratification was led by in-country staff did not indicate challenges with meeting length or partner communication in the same way.

Participants reported particularly low clarity around the mathematical modeling portion of SNT. Participants expressed that although there was a growing interest in using modeling, their limited understanding of modeling made it difficult to engage with modeling outputs and their implications. Despite thorough and frequent conversations with analysis partners involved in mathematical modeling, most NMPs reported not understanding the modeling outputs:

*No, in fact, we didn't understand the [mathematical] modeling. We know the interventions that were explained to us. If we're going to do interventions with coverage at such and such a level, that's what we can estimate; but if the coverage remains standard, here's what we can save, that we can reduce the gap. And so on. These are the results we've been told. But how were these results generated? Well, we didn't understand that.* (Participant 11)

One NMP was particularly skeptical of mathematical modeling and vigilant in keeping a close watch on what went into the models:

*We have always been a bit skeptical about [mathematical] modeling because modeling is basically based on what you put in, then you get the results. We are careful in the initial phases about what was being put in. We have to basically be close with them in the final stages of the modeling so that we are sure of what will come out of the modeling. Otherwise later we may have issues.* (Participant 4)

Several participants mentioned that working with an in-country partner with a strong understanding of mathematical modeling would have provided more clarity and opportunities for detailed explanations of modeling outputs. Two NMPs felt that it would be helpful to have documentation that described each step of the mathematical modeling process in a way that was accessible to the NMP. Participants across all NMPs were optimistic that their understanding of modeling outputs would grow with time, similar to their growing understanding of other SNT outputs.

### Ownership

Participants were asked directly about their sense of ownership of the SNT analysis process and to provide examples of how their autonomy was supported or challenged throughout the process. Their responses resulted in NMP-defined themes of ownership that included general ownership of the overall process, autonomy over the use of technical outputs, and NMP capacity to advocate for themselves in the face of disagreements with analysis or implementing partners. Participants reported feeling comfortable explaining the SNT results and using them in diverse contexts, which contributed to their growing sense of ownership as they gained more experience with SNT.

Participants reported that the SNT analysis process was highly participatory and genuinely NMP-led, which, during Phase 1, was counter to expectations for some NMPs. NMPs who initially felt obliged to participate in SNT to appease donors, WHO, and/or prospective partners reported a low sense of ownership during the start of Phase 1, with low expectations of their autonomy throughout the process. However, their sense of ownership grew over the course of Phase 1 and was strong during Phase 2:

*They [WHO] made us aware of our responsibilities very early on. Well, they told us, at their level, they only produce models, it's up to us to validate them. Afterward, it's up to us to implement them or not. That was clear from the start. Even with us, it was clear from the very beginning. I admit that, at first, I didn't believe it. [...] So, in the beginning, I was very reluctant. I mean, I was informed that I was performing the task, but it was during the process that I began to realize how much I could actually benefit from it.* (Participant 7)

The NMP that only participated in Phase 2 reported a high sense of ownership throughout, beginning with initiation. Participants from this NMP felt that they were driving SNT implementation and directed committees comprised of technical partners, NGOs, health faculty, and other ministry partners. Participants from this NMP defined their sense of ownership through their increased capacity to implement epidemiological stratification and their ability to provide critical feedback about technical outputs.

Participants cited analysis partner commitment and prioritization of NMP needs as primary enabling factors for ownership. It was felt that analysis partners' willingness to engage in iterative meetings and pace the discussion at the NMP's

level of comfort reflected the partners' deep commitment to supporting the NMP in meeting its objectives. Participants reported giving regular feedback to analysis partners and validating outputs at each stage of analysis, beginning from the early steps of the SNT process. In situations where the NMP was unable to carry out any of the analysis work on their own, NMPs still felt confident in providing their expertise of the local context and comfortable asking for clarification on the analysis approach and outputs:

> *He [analysis partner] was also available when we asked questions for clarification. And no, they never, never complained about us asking too many questions. On the contrary, they encouraged us to take more. If ever there were things, concerns about judgment, clarification, if we didn't get the answers now, or if time didn't allow it, to have certain elements [discussed] at the next call, we didn't hesitate to come back to it, to ask to have it explained further to us.* (Participant 3)

Participants indicated that the space to openly critique analysis outputs was important to them. As analysis partners began providing outputs for NMPs to review, participants reported that they were able to provide critical feedback and correct any inaccuracies they observed. Participants provided several examples where analysis partners were responsive to feedback and corrected outputs to reflect on-the-ground realities more accurately. In one situation, a stratification presentation inaccurately indicated that health facilities in a specific region were closed. Following discussions, analysis partners were quick to ensure that the status of health facilities was accurately reflected in subsequent analyses. In another situation, rainfall from a global database was included in the mathematical model because local rainfall data was not available. NMP members pointed out that the resulting seasonality in the model did not match their experience. The analysis partners then removed rainfall from the model, which led to model outputs that the NMP found to be consistent with what they knew of local epidemiology:

> *I noticed one of the models. I really challenged it. [Analysis partner] who was there, understood. So, I think that even before coming to the meeting, [analysis partner] was on my idea. So, after a while of discussion, [analysis partner] said, well, we're going to remove that component from the interior [of the model].* (Participant 7)

According to participants, their close involvement in the analysis process helped foster deeper comprehension of how analysis partners arrived at their conclusions. This enabled participants to communicate the implications of different outputs with implementing partners and interministerial partners more effectively:

> *I think the strong point of the collaboration we've had with the WHO and its technical support for this exercise is that we've been involved in it. The fact that we had several meetings, around ten meetings, so phases that were organized on a weekly basis, where all the [program] staff and its implementing partners who had a very good knowledge of malaria transmission, of the local context, really enabled us to accept the results. So we were able to take ownership of the results, defend them and use the right language to defend our work.* (Participant 6)

Particularly in Phase 2, NMPs felt comfortable adapting the SNT process to suit their own needs. At least one interviewee from each NMP drew strong connections between the NMP's close work with analysis partners and the NMP's growing capacity to implement parts of the SNT analysis on their own. Close involvement in the analysis steps of SNT, coupled with greater confidence in communicating outputs to other stakeholders, laid the foundation for NMPs' capacity to implement SNT analysis without external support. One participant described how NMP capacity and NMP ownership are intertwined, particularly as the SNT process requires adaptation to a changing local context:

> *We learn as we implement and these things we do, they're not cast in stone. They are living things. They tend to change as we move along. So something that is so like "it has to be this number 1 to 5 steps." No. You would need to understand*

*what works best at the time. As we are implementing, we are getting to understand how best we can do this thing better. [... So the] capacity part of things is very important and that ownership from the country is also very, very important. And which you can see the relation between the capacity and the ownership come side-by-side.* (Participant 9)

**Future plans for SNT**

All participants stated explicit plans for their NMPs to continue using and even expanding SNT, citing its value and the lessons learned for improving prevention and care outcomes thus far. Several participants wanted to extend SNT to even finer geographical levels:

*We're going to continue [with SNT] and even go further. As we move forward in the fight against the disease, we need to be more and more specific. So, we may need to go to the health facility level to see, at district level, which health facilities have a high incidence, which health facilities are like the ones we have now, what do we need to do to target these health facilities in order to make progress? Because we need to put ourselves in the position of eliminating malaria.* (Participant 11)

**Plans for new resource mobilization approaches**

Following the successful use of SNT outputs for Global Fund applications, NMPs began to explore how they could reuse SNT outputs for other donors, particularly the private sector. Participants described plans to seek resources from private sector malaria elimination initiatives to fill funding gaps. Three NMPs identified funding opportunities from private sector partners and initiated discussions using SNT outputs to demonstrate their proposed activities and funding justifications. At the time of the interview, one NMP was in the process of establishing a private sector committee to eradicate malaria, drawing in private sector partners across various industries. This committee was specifically designed to mobilize funding for the national strategic plan, built on SNT outputs, which had yet to be implemented because of limited funding.

**Plans for data quality improvement and subnational-level engagement**

Participants spoke extensively about their plans for improvements in data management and data quality through improved documentation, centralized data repositories, and, most importantly, systematic engagement of subnational offices and health facilities. Participants emphasized the role that subnational offices could play in addressing data quality issues in a timelier fashion and improving the acceptability of outputs. In their view, health facility staff are key players in malaria elimination efforts and must be included more intentionally:

*If I had the opportunity to do this exercise again, I would insist that the actors responsible for the health districts take part in this process. [...] Given that they are the ones directly responsible for combating the disease at operational level, their involvement will be very useful because it will enable them to appropriate this technique and to have the necessary resources and skills to respond in the event of the identification of a certain number of anomalies. So we need to involve the operational level players, who will help to facilitate and complete the analysis […] and validate the assumptions. But they can also use this tool [SNT] for planning purposes, in the context of stratification at the district level, for example.* (Participant 6)

**Plans for developing the capacity to implement SNT analyses independently**

Most NMPs reported high confidence in conducting stratification on their own but felt that they would need additional support for microstratification and mathematical modeling. The NMPs with low confidence in managing stratification on

their own reported that while the SNT process had been inclusive, their involvement was not enough to understand how to execute the analyses themselves:

*As our participation was passive and not active, since we just provided the data, and attended meetings, we won't be able to do it without capacity building. There is a process involved. Generally, people use software for data collection. However, it is not easy to master the [analysis] software. Someone must explain to us how it works.* (Participant 2)

While engaging a local expert would allow NMPs to have more in-person support, participants highlighted that this might compromise timelines if the consultant is engaged with other projects. For one NMP, their experience with local consultants was such that when the consultant had a more lucrative contract, the NMP's work was deprioritized.

Some participants felt that support from WHO and external analysis partners strengthened NMPs' legitimacy in the eyes of prospective donors and other partners. As NMPs navigated the power dynamics involved with funding, they were cognizant of the persistent devaluation of local expertise, even when they felt that their expertise was being valued in their collaborations with analysis partners. NMPs expressed concerns about whether the acceptability of their outputs or competitiveness for funding would change without the external validation of international institutions, highlighting the need for a delicate balance between legitimacy and autonomy. Participants with high confidence in executing analyses on their own felt that this balance could be achieved if the NMP performed most of the work and sent their work to WHO or analysis partners to receive their endorsement. Executing analysis themselves would also let the NMP engage more deeply and effectively with their partners:

*It's true that at a certain point, well, we extracted the data, transmitted it and waited. As a result, we didn't feel much of a player in the field, and we didn't have the capacity to use it either. [...] We hope to have a better capacity, at the level of the [program], to not only manage the data, but of course to have the support and possibilities of a more correct exchange with [analysis partner]. So when we've already done some analysis, we'd like to have a different interpretation and/or more in-depth guidance.* (Participant 3)

Regardless of confidence in their current capacity for implementing some or most SNT analyses internally, all participants reported a deep desire to strengthen NMP capacity for analysis. One participant hoped to have more than one focal person working with analysis partners to better mitigate timeline challenges and knowledge loss from NMP turnover. Some participants felt that more emphasis should have been placed on capacity development during the SNT process to better ensure sustainability:

*When countries are doing most of the work, rather than now coming from outside, will also add a lot more value. The feedback I would say to WHO is identify those that would lead the process in-country and then build capacity to some extent that they will steer the process with little or no support from the external. Because if you try to say "let me do everything for them because they are just starting," you end up not getting them to the speed that you require, even if they are going to do it again. So let them concentrate more in building capacity. Let every other person that is required to know, that this is how modeling is being done. This is why we are doing it. If it's going to take longer time, wait for that time to come to build necessary capacity before you run to carry out the SNT process in countries. It's better to delay [than to] rush and do it while the countries are not on board because they understand little or nothing in the process.* (Participant 9)

## Discussion

This study centered on NMPs' self-reported perspectives to understand their experiences with the SNT process they underwent in partnership with WHO and collaborators. This study found that positive experiences with the SNT process

enabled NMPs to appropriately target interventions, refine strategic plans, optimize and prioritize resources, and identify opportunities to strengthen data systems and capacity for data use. This study also found that NMPs felt a strong sense of ownership of the SNT process, effectively advocated for themselves in the face of competing partner priorities, and were invested in continuing to use and improve SNT in the future. Participants' testimonies also shed light on the different ways in which external parties can support the SNT process while safeguarding NMPs' ownership and autonomy.

### Acceptability and use of SNT outputs

All NMPs included in this study used SNT outputs to inform intervention planning in either their national strategic plans, funding requests, or both. Among the decisions made using SNT outputs, NMPs referred most commonly to decisions around the identification of highest-risk areas, expansion of SMC, and targeting of LLIN types, the major prevention interventions currently implemented in malaria-endemic countries. Other countries not included in this study have also used SNT-like approaches to inform these decisions [14,16,19]. NMPs' incorporation of SNT outputs directly in their strategic plans demonstrated the validity and acceptability of the results of the SNT process to NMPs, even if the full strategy could not be included in funding requests to donors, given resource gaps and other constraints. The acceptability of SNT outputs was also evident in NMPs' commitment to improving acceptability at the subnational level through more engagement of subnational offices, as well as countries' willingness to use SNT outputs to mobilize additional resources, such as from the local private sector or domestically. This finding further demonstrates that private sector engagement is a critical yet underutilized opportunity for NMPs to procure the additional funding needed to support intervention implementation [21].

### Shifts in perspectives on data and data use

Implementation of SNT drove shifts in NMP perspectives about data, including its use, management, and quality improvement. NMPs understood more deeply the strengths and limitations of their current data collection and became more invested in maximizing their data use to inform decisions. NMPs recognized that many issues in data gaps and quality that were identified during SNT were partly due to a lack of training on data collection and use at the health facility level, as has been reported in other studies [22,23]. Other studies have also found that using data to inform decision-making in malaria highlights the limitations of routine data and the need for improved data quality management [14,15,21,22]. This study found that SNT motivated NMPs to strengthen surveillance, create integrated data repositories, stratify at finer geographical levels, modify their approaches to data coordination, and strengthen capacity for data use at all levels of the health system.

One way in which SNT resulted in a growth in the culture of data use is illustrated in the multiple ideas presented by participants on stratification at smaller geographic units, which would better represent the heterogeneity of malaria transmission and its determinants, especially for countries with sizable geographic and population diversity [24]. More detailed stratification would require data quality improvement through the engagement of subnational offices. This revealed the potential cascading effects on data use that the SNT experience could have beyond the central office of the national malaria program. As data quality improves, the perceived robustness of the analytical results should also improve, with both factors acting together to boost the confidence of NMPs to make evidence-informed decisions with their data.

### Capacity for conducting SNT

All participants alluded to the different levels of capacities available in their NMP, as well as the current capacity gaps, with respect to what would be needed for their NMP to completely, independently, and locally conduct an SNT process. Multiple capacities would be required, including leadership in policy-making and coordination, data coordination and management, basic descriptive analysis, advanced analytics such as geostatistics and modeling, and health economics and budget management. These skill sets can be shared by different individuals and institutions, both within and outside government. Approaches for capacitating should be adapted across skills, skill levels, and audience, both at national and

subnational levels. The growth of these capacities is essential for countries to continue to expand and adapt SNT-like approaches.

Findings from this study suggest that funders and partners interested in sustainable evidence-informed decision-making should focus on allocating resources and efforts to deliberately develop capacities where there are gaps, which can differ substantially between countries. As capacity grows, funders also need to develop trust in that capacity and the locally-derived technical outputs it produces without the need for validation from external institutions, as some participants pointed out in the study.

### Ownership

This study arrived at definitions of country ownership defined by each participant. Others have noted that only localized, context-specific definitions of ownership can determine when and how country ownership is being realized [25]. Context-specific definitions enable better measurement of the power dynamics in the "daily practice" of donor-recipient relations, including the extent to which local expertise is valued and involved at all stages [26]. In the case of SNT, the exploration of ownership found that open communication, strong collaboration, and decision-making power were the key components that contributed to NMP members' definitions and sense of ownership.

Although participants felt ownership of the SNT process, they still experienced challenges leveraging SNT outputs to convince implementing partners and other external stakeholders to align with NMP priorities. Other studies of equity in global health collaborations found that although national stakeholders note a shift in efforts around collaborative goal-setting, they are still cognizant of underlying power dynamics and proactively adapt to donor priorities [27]. NMPs' keen awareness of the existing power dynamics between themselves, funders, implementers, and other partners is also evident in the findings from this study.

### Enabling factors for analysis partners

This study found that proactive inclusion, extensive dialogue, trust-building, and adaptability on the part of technical partners were important factors in enabling countries to have positive experiences with the SNT analysis process. The enabling factors identified in this study align with previous research where national stakeholders in Nigeria and the Democratic Republic of the Congo characterized effective technical assistance as country-driven, cultivating self-reliance and autonomy, trust-focused, and adaptive [28]. Despite the reported availability of SNT technical partners to answer questions, engagements were still insufficient for NMPs to understand more complex methods, such as mathematical modeling. This speaks to the need for technical partners to spend even more time explaining complex methods inclusively and accessibly so that NMPs can better understand how and when to use these methods and interpret their outputs appropriately.

### Implications for the HBHI approach and priority-setting in public health

Among the NMPs included in this study, SNT was a highly successful approach for the strategic use of information in decision-making, including for prioritization under resource constraints. The impact of these decisions on malaria burden, including whether stagnating trends started decreasing, will be seen in the coming years. However, the SNT exercises of 2019–2023 have already spurred changes within NMPs, where there is strong momentum to continue adapting SNT techniques to tailor intervention strategies to the local context and advocate for country priorities. Beyond malaria, ministries of health, partners, and donors can learn from NMPs' experiences with SNT to design effective, country-owned, and sustainable initiatives to improve evidence-informed decision-making in other parts of the health system.

### Limitations

This study is limited by its sample size and participant selection method. Although all NMPs who were supported by WHO to implement the full SNT analysis during Phase 2 were asked to participate in this study, one NMP was not responsive

to requests. It is therefore possible that the inclusion of additional NMPs could have identified other challenges or opportunities for improvement. Additionally, the findings of this study are geographically limited to the experience of NMPs from West Africa.

This study's findings also may not apply to countries that underwent SNT without WHO involvement. During Phase 2, several countries underwent SNT either on their own or with the technical support of partners other than WHO, and the process may have differed. Their experiences of ownership, use of outputs, and relationships with partners may not have been the same as those reported by the countries included in this study.

## Conclusions

NMPs reported positive experiences with the SNT process and a growing appreciation for the utility of SNT for malaria intervention planning and resource mobilization. SNT outputs were widely used and highly acceptable. Despite challenges with data quality, availability of local capacity, and difficult decisions under limited resources, NMPs reported strong enthusiasm for the continued use of SNT, articulated plans to extend it in new ways, and expressed a desire to grow capacity to conduct analyses on their own. Trust-building, iterative discussions, and NMP-centered approaches were instrumental to achieving the results observed.

## Supporting information

**S1 File. Informed consent form: Qualitative assessment of NMCPs' perceptions of the stratification and sub-national tailoring of interventions analytical processes.**
(DOCX)

**S2 File. Interview gsuide: Qualitative assessment of NMCPs' perceptions of the stratification and sub-national tailoring of interventions analytical processes.**
(DOCX)

## Acknowledgments

The authors would like to thank Mwalenga Nghipumbwa, Laurent Moyenga, and Yacouba Savadogo for assisting with mock interviews during study design, and Abdisalan Noor for helpful discussions.

*Disclaimer:* Beatriz Galatas is a staff member of the World Health Organization. This author alone is responsible for the views expressed in this article and does not necessarily represent the decisions, policies or views of the World Health Organization.

## Author contributions

**Conceptualization:** Letitia Onyango, Ifeoma Ozodiegwu, Beatriz Galatas, Jaline Gerardin.

**Data curation:** Letitia Onyango, Ghislaine Ouédraogo-Ametchie.

**Formal analysis:** Letitia Onyango, Ghislaine Ouédraogo-Ametchie, Beatriz Galatas, Jaline Gerardin.

**Investigation:** Letitia Onyango, Ghislaine Ouédraogo-Ametchie.

**Methodology:** Letitia Onyango, Ghislaine Ouédraogo-Ametchie.

**Writing – original draft:** Letitia Onyango, Beatriz Galatas, Jaline Gerardin.

**Writing – review & editing:** Letitia Onyango, Ifeoma Ozodiegwu, Beatriz Galatas, Jaline Gerardin.

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
