## [Decision Letter · Decision Letter 0]

9 Feb 2025

PGPH-D-24-02161

Subnational tailoring of malaria interventions for strategic planning and prioritization: experience and perspectives of five malaria programs

Dear Dr. Gerardin,

Thank you for submitting your manuscript to PLOS Global Public Health. After careful consideration, we feel that it has merit but does not fully meet PLOS Global Public Health’s publication criteria as it currently stands. Therefore, we invite you to submit a revised version of the manuscript that addresses the points raised during the review process.

The manuscript has been evaluated by four reviewers, and their comments are available below.

The reviewers have raised a number of major concerns. In particular, they feel the manuscript should outline a clear and persuasive rationale for the study, and they request improvements to the reporting of methodological aspects of the study.

Could you please carefully revise the manuscript to address all comments raised?

We look forward to receiving your revised manuscript.

Kind regards,

Helen Howard

Staff Editor

Journal Requirements:

2. Please make sure the funding information on the submission form matches your financial disclosure statement. Please indicate by return the full and correct funding information for your study and confirm the order in which funding contributions should appear. Please be sure to indicate whether the funders played any role in the study design, data collection and analysis, decision to publish, or preparation of the manuscript.

3. In the online submission form, you indicated that "Given the small sample, and identifiability of participants, the full transcripts cannot be provided. Codebooks and aggregated, anonymized analyses will be available upon request.". 

3. Uploaded as supplementary information.

4. We note that "supplemental file 1.docx" should not be published with the paper and are for internal information only. Please changed the file type to 'Other' on your behalf to ensure that they are not published. 

Additional Editor Comments (if provided):

Reviewers' comments:

Reviewer's Responses to Questions

**Comments to the Author**

1. Does this manuscript meet PLOS Global Public Health’s publication criteria? Is the manuscript technically sound, and do the data support the conclusions? The manuscript must describe methodologically and ethically rigorous research with conclusions that are appropriately drawn based on the data presented.

Reviewer #1: Yes

Reviewer #2: Yes

Reviewer #3: Yes

Reviewer #4: Yes

2. Has the statistical analysis been performed appropriately and rigorously?

Reviewer #1: N/A

Reviewer #2: N/A

Reviewer #3: N/A

Reviewer #4: Yes

3. Have the authors made all data underlying the findings in their manuscript fully available (please refer to the Data Availability Statement at the start of the manuscript PDF file)?

Reviewer #1: No

Reviewer #2: No

Reviewer #3: Yes

Reviewer #4: Yes

4. Is the manuscript presented in an intelligible fashion and written in standard English?

Reviewer #1: Yes

Reviewer #2: Yes

Reviewer #3: Yes

Reviewer #4: Yes

5. Review Comments to the Author

Reviewer #1: Congratulations on a really interesting and valuable paper.

On my first read I considered if some sections could be reduced since the paper is quite long, however given the limited other literature providing similar feedback on this topic and the comprehensive approach that you have taken, I think the length is justified.

I have no other edits to suggest, and recommend the paper be accepted.

Reviewer #2: The study provides an interesting and useful analysis of the experience of participants in the SNT process, which is highly topical given WHO’s ongoing development of an SNT implementation manual. Of particular interest (to me at least) are the reported challenges relating to subnational partners and the need for improved in-country mathematical modelling expertise.

As a health economist with limited experience in qualitative research methods, I have primarily focused on how the paper describes the SNT process and its application for efficient priority-setting.

MAJOR COMMENTS

High-level comments mostly relate to the framing of the study and the rationale for SNT.

1. The Introduction notes several factors contributing to stalled progress in reducing malaria morbidity and mortality (lines 55-57) but does not clearly explain why, nor really get to the underlying drivers of this stalled progress. For example, “sub-optimal malaria intervention coverage” is mentioned as a factor; this is due to a stagnation in global funding for malaria, and the increasing difficulty (and cost) of expanding coverage to harder-to-reach-populations. The impact of insecticide and antimalarial resistance should also be explicitly mentioned; such resistance has reduced intervention effectiveness and required the adoption of more expensive alternatives.

2. The origin of SNT as part of the HBHI approach is described (although this is very lightly referenced, particularly lines 67-107). However, the underlying rationale for SNT is not clearly explained. The effectiveness and cost-effectiveness of interventions varies by geographic area and population, and hence a more efficient allocation of resources is achieved by tailoring intervention delivery based on the stratification of various components of malaria risk. Heterogeneity in malaria risk (which is not mentioned until the Discussion section) underpins the rationale for SNT, and is increasing as incidence declines in many areas. For further detail, refer to (and consider citing) WHO’s recent updates on SNT: ‘Update on subnational tailoring of malaria interventions and strategies’ (Malaria Policy Advisory Group Meeting, 4-5 March 2024) and ‘Guiding principles for prioritizing malaria interventions in resource constrained country contexts to achieve maximum impact’ (May 2024).

3. The Introduction could also provide a clearer rationale for the importance of the study, including the benefits of an improved understanding of participant experiences of the SNT process.

MINOR COMMENTS

1. Abstract: Consistent with the major comments, the abstract should also provide further background on why SNT is important in order to maximise the health impact of scarce resources across heterogenous areas and populations.

2. Line 44: Chemoprevention is mentioned, however this is not supported by the reference provided (Bhatt et al).

3. Lines 208-9: Were the interviews transcribed and translated by members of the research team?

4. Lines 415-32: The analysis relating to “prioritization … to account for costs” is not clearly explained. From the quotes it appears that activities that were prioritized as part of the SNT process were later excluded due to their cost. These raises an important issue regarding the SNT process: prioritisation is currently done based on effectiveness and then subsequently costed, rather than prioritising based on cost-effectiveness – which would take into account heterogeneity of costs and enable more efficient resource allocation.

5. Line 916: This is the first time heterogeneity is mentioned, although it is central to the rationale for SNT. Note that heterogeneity is not just in terms of malaria transmission, but many other factors of malaria risk (including insecticide and drug resistance, access to care, etc).

6. Line 977: This section could refer to efficiency more explicitly – i.e. improving the health impact of limited malaria control resources.

Reviewer #3: I thought this paper was well written and an excellent complement to published examples of the SNT outputs. However, I do think some of those concrete examples of use of SNT outputs could be reiterated here rather than just referenced. These concrete examples would bolster the positive claims of the qualitative interviews.

Specific comments:

ln. 57: insecticide resistance was likely a factor here too for both nets and IRS

ln 109 and 116: this paragraph might be well illustrated with a table to show countries/types of assistance, etc.

ln 134: examples of SNT use. Several of the articles referenced were published too early to have been the products of the SNT process described here (although they were using subnational data to make decisions). However, for a few of the later publications, there are concrete examples of how SNT changed the status quo...it would be very useful to have a few of these concrete examples in this article.  

ln 885-888: can you give concrete examples of decisions made using SNT that might not have happened in the absence of the SNT process. Some of the papers in the references contain actual examples and perhaps some of them could be cited here.

References: some look like preprints (not peer reviewed). If the paper has subsequently appeared after peer-review, please update the reference

Reviewer #4: The manuscript presents a qualitative study exploring the perspectives of National Malaria Programs (NMPs) on the use of Subnational Tailoring (SNT) for malaria strategic planning and prioritisation. While the study is valuable in documenting experiences across five countries, several areas require improvement to enhance clarity, strengthen the analytical depth, and improve the overall presentation.

Major Comments

1. Conceptual Clarity and Rationale for SNT Approach

• The manuscript describes SNT as a novel approach, but does not clearly differentiate it from existing malaria stratification methods used in endemic countries.

o How does SNT differ from traditional malaria stratification approaches used by WHO or NMCPs?

o What are the specific advantages of SNT over conventional methods in terms of decision-making?

o Was there a preceding malaria stratification framework in these countries before adopting SNT?

o Suggested Revision: Provide a clearer theoretical grounding for SNT and its distinct value-add.

2. Overuse of WHO-Centric Perspective

• The WHO's role in SNT implementation is strongly emphasized, but the manuscript lacks a critical discussion on:

o How NMPs perceived WHO’s role—was it more of a technical enabler or a decision-making authority?

o Did WHO’s involvement introduce limitations, such as prescriptive frameworks that may not fully align with country-specific needs?

o Suggested Addition: Provide a balanced discussion on both facilitating and limiting aspects of WHO’s involvement.

3. Participant Selection and Generalizability

• The study relies on interviews with 12 NMP representatives from five countries. However:

o Why were only NMP representatives included? Were perspectives from other stakeholders (e.g., funders, health workers, community leaders) considered?

o Suggested Revision: Acknowledge the limited scope and potential biases introduced by focusing only on NMP-level perspectives.

4. Limited Cross-Country Comparison

• The manuscript discusses SNT implementation country by country, but lacks an effective cross-country synthesis.

o What are the key similarities and differences in how each country implemented SNT?

o Are there country-specific barriers that affected implementation differently (e.g., governance, funding constraints, health system structure)?

o Suggested Revision: Introduce a comparative analysis table summarizing differences in implementation approach, challenges, and perceived benefits.

5. Impact Assessment of SNT on Malaria Outcomes

• The manuscript reports that SNT led to improved strategic planning, but provides limited evidence of its direct impact on malaria burden reduction.

o Are there any measurable improvements in malaria incidence, intervention coverage, or funding acquisition attributable to SNT?

o Suggested Revision: Include specific data (even preliminary) on how SNT changed decision-making and improved malaria program outcomes.

6. Discussion and Recommendations

• The discussion lacks concrete takeaways for malaria program managers and policymakers.

o What specific lessons can other malaria-endemic countries learn from this study?

o What are the actionable steps for countries considering SNT implementation?

o Suggested Addition: Provide a structured set of recommendations on how SNT can be operationalized in different malaria-endemic contexts.

Minor Comments

1. Language and Readability

• The manuscript contains excessively long and complex sentences that make it difficult to read.

o Example:

"Participants expressed that SNT outputs were used to inform national strategic plans and prioritized plans, that the process incentivized improvements in data collection and data quality, and that NMPs were strongly motivated to grow their capacity to conduct more steps of the SNT analysis process internally."

o Suggested Revision:

"Participants reported that SNT helped shape national strategic plans, improved data quality, and motivated NMPs to develop internal analytical capacity."

2. Overuse of Acronyms

• The manuscript frequently uses SNT, NMP, WHO, HBHI, IRS, SMC, ACTs, often without proper reintroduction in later sections.

o Suggested Fix: Provide a glossary of key terms and acronyms.

3. Abstract Needs Refinement

• The abstract repeats details already present in the main text and could be more concise and impactful.

o Suggested Fix: Reduce methodological details in the abstract and focus more on key findings and policy implications.

4. Figure and Table Formatting

• Table captions are too generic and do not sufficiently explain what the data represents.

o Example: Instead of "Table 1: Summary of SNT implementation steps", use "Table 1: Steps followed in the subnational tailoring (SNT) process across five malaria programs"

• Figures (especially maps) lack clarity on geographic regions analyzed.

o Suggested Fix: Provide more precise labels and explanations.

Final Verdict: Major Revision

The manuscript is well-structured but lacks analytical depth and clear cross-country comparisons. Key areas requiring revision include improving the conceptual framework, providing stronger evidence of impact, refining the discussion, and making the language more concise. Once these issues are addressed, the manuscript will make a valuable contribution to malaria program planning and implementation research.

6. PLOS authors have the option to publish the peer review history of their article (what does this mean?). If published, this will include your full peer review and any attached files.

**Do you want your identity to be public for this peer review?** For information about this choice, including consent withdrawal, please see our Privacy Policy.

Reviewer #1: No

Reviewer #2: No

Reviewer #3: No

Reviewer #4: **Yes: **Harsh Rajvanshi

---

## [Decision Letter · Decision Letter 1]

15 Apr 2025

Subnational tailoring of malaria interventions for strategic planning and prioritization: experience and perspectives of five malaria programs

PGPH-D-24-02161R1

Dear Dr. Gerardin,

We are pleased to inform you that your manuscript 'Subnational tailoring of malaria interventions for strategic planning and prioritization: experience and perspectives of five malaria programs' has been provisionally accepted for publication in PLOS Global Public Health.

Best regards,

Ruth Ashton, Ph.D.

Academic Editor

Reviewer Comments (if any, and for reference):

Reviewer's Responses to Questions

**Comments to the Author**

1. If the authors have adequately addressed your comments raised in a previous round of review and you feel that this manuscript is now acceptable for publication, you may indicate that here to bypass the “Comments to the Author” section, enter your conflict of interest statement in the “Confidential to Editor” section, and submit your "Accept" recommendation.

Reviewer #3: All comments have been addressed

2. Does this manuscript meet PLOS Global Public Health’s publication criteria? Is the manuscript technically sound, and do the data support the conclusions? The manuscript must describe methodologically and ethically rigorous research with conclusions that are appropriately drawn based on the data presented.

Reviewer #3: Yes

3. Has the statistical analysis been performed appropriately and rigorously?

Reviewer #3: N/A

4. Have the authors made all data underlying the findings in their manuscript fully available (please refer to the Data Availability Statement at the start of the manuscript PDF file)?

Reviewer #3: Yes

5. Is the manuscript presented in an intelligible fashion and written in standard English?

Reviewer #3: Yes

6. Review Comments to the Author

Reviewer #3: This is an excellent qualitative study examining the perspectives of malaria control programs regarding the HBHI initiative. It is well written and addresses many of the underlying issues of local acceptability of the initiative. Kudos to the authors for actually collecting and analyzing the qualitative data to support the quantitative work of HBHI.

7. PLOS authors have the option to publish the peer review history of their article (what does this mean?). If published, this will include your full peer review and any attached files.

**Do you want your identity to be public for this peer review?** For information about this choice, including consent withdrawal, please see our Privacy Policy.

Reviewer #3: No
